# Storage Conditions of Sperm Samples and Gametic Characterization by Sperm Head Morphometry in Drones (*Apis mellifera*)

**DOI:** 10.3390/ani15050672

**Published:** 2025-02-26

**Authors:** Milagros Cristina Esteso, Adolfo Toledano-Díaz, Cristina Castaño, Mariano Higes, Raquel Martín-Hernández, Agustin López-Goya, Pilar De la Rúa, Belén Martínez-Madrid, Julián Santiago-Moreno

**Affiliations:** 1Departamento de Reproducción Animal, INIA-CSIC, 28040 Madrid, Spain; milaesteso542@gmail.com (M.C.E.); cristina.castano@inia.csic.es (C.C.); moreno@inia.csic.es (J.S.-M.); 2Laboratorio de Patología Apícola, Centro de Investigación Apícola y Agroambiental, CIAPA-IRIAF, Consejería de Agricultura de la Junta de Comunidades de Castilla-La Mancha, 19180 Marchamalo, Spain; mhiges@jccm.es (M.H.); rmhernandez@jccm.es (R.M.-H.); 3FAUNIA, 28032 Madrid, Spain; algoya@grpr.com; 4Área de Biología Animal, Departamento de Zoología y Antropología Física, Universidad de Murcia, 30100 Murcia, Spain; pdelarua@um.es; 5Departamento de Medicina y Cirugía Animal, Facultad de Veterinaria, Universidad Complutense de Madrid, 28040 Madrid, Spain; belmart@ucm.es

**Keywords:** short-term storage, seminal quality, catalase, sperm morphometry, honey bees

## Abstract

Environmental factors may produce morphological and morphometric alterations in the honey bee’s sperm. These changes in drone spermatozoa can produce variations in their physical characteristics, which could cause the loss of ability to remain viable within the queen bee’s spermatheca. The storage and conservation of sperm is an effective strategy to protect the genetic diversity of honey bees and contribute to selective breeding programs. The extender choice greatly impacts sperm quality. The use of antioxidants in extenders for preservation could improve the functional semen characteristics. Indeed, the supplementation of extender with catalase (200 UI) improved the sperm viability and motility during liquid storage. The accurately measured dimensions of sperm head allow improving the understanding of reproductive biology and might even be used as indicators of environmental influences. The variability of sperm head morphometry was calculated, revealing a high degree of sperm pleomorphism in drones, suggesting the existence of sub-populations that should be investigated in the future.

## 1. Introduction

Honey bees are important pollinators of agricultural and non-agricultural landscapes. Declines in insect pollinator populations and substantial losses in managed honey bees have been reported on a global scale and become a widespread concern because of the importance of these insects for human food production and ecosystem stability. Several potential factors have been studied as possible causes of declining pollinator health, such as parasites and pathogens, exposure to agricultural pesticides, habitat loss and/or climate change [1,2].

Honey bee colonies are very vulnerable to the fertility of queen bees. The mating process initiates numerous behavioral, physiological, and molecular changes that may affect the queen’s fertility and her influence on the colony. Queens mate with multiple drones during a single mating period early in life, where they obtain enough sperm to fertilize their eggs for the rest of their reproductive life [3]. Pesticides derived from agricultural and beekeeping activity affect the immune system, the nervous system, the orientation system, and the reproductive system of the bees [4,5]. In this sense, it has been shown that these pesticides produce sub-chronic or chronic damage, mainly affecting cells with high mitotic rates, such as male germ cells. Toxic substances affect spermatogenesis and sperm functionality in amphibians [6], birds [7], and mammals [8]. Moreover, climate change produces morphological and morphometric alterations in sperm. Changes in the morphology of drone spermatozoa can produce variations in their physical characteristics, which could cause the loss of ability to remain viable within the queen bee spermatheca. If this occurs, the genetic diversity within the hives would be endangered, preventing the proper fulfillment of the tasks necessary for the proper functioning of colonies [9]. To solve possible reproductive problems in drones, we must first identify the appearance of gametes abnormalities and alterations in the semen functionality [10,11].

Since the 1960s, the influence of different extenders and storage temperatures on drone semen has been investigated [12]. The storage and conservation of sperm is an effective strategy to protect the genetic diversity of honey bees and contribute to selective breeding programs [13]. The extender choice greatly impacts sperm quality [14]. Since there is often a delay between semen collection and analysis, the use of a suitable extender becomes a priority. There is evidence indicating that sperm cell dysfunction is induced by oxidative stress [15]. The use of antioxidants in extenders for preservation and improvement of sperm characteristics in domestic [16,17,18,19] and wild species [20,21,22] has been widely studied. The enzymatic antioxidant function is to suppress the formation of free radicals by breaking them down [23]. In this sense, catalase comprises several protein subunits, each with a heme group and a nicotinamide adenine dinucleotide phosphate (NADPH) binding site. Its function is to prevent damage induced by reactive oxygen species (ROS) by activating hydrogen peroxide (H_2_O_2_) and obtaining simple molecules of water (H_2_O) and oxygen (O_2_) [24]. It is proven that the supplementation of extenders with catalase improves sperm parameters in several species [25,26,27]. Thus, we aimed to check if the same happens in drones.

Looking into all this information, the present study aimed to evaluate an optimal method to transport and store drone sperm samples, as well as to characterize drone spermatozoa through sperm head morphometry.

## 2. Materials and Methods

### 2.1. Experimental Design

A total of 291 drones (*Apis mellifera iberiensis*) from three different places were used in this study: Faculty of Veterinary Medicine of University from Murcia (UMU) (Latitude 38°01′35.4″ N, longitude 1°10′33.1″ W. Altitude 150 m), Murcia, Spain; Apiculture and Agro-environmental Research Center (CIAPA-IRIAF) (Latitude 40°40′48.54″ N, Longitude 3°12′30.05″ W. Altitude 670 m), Marchamalo, Guadalajara, Spain; Faunistic Theme Park-FAUNIA (Latitude 40°23′36.5″ N Longitude 3°36′51.3″ W Altitude 680 m), Madrid, Spain. The hives of UMU were located in a semi-urban area on the outskirts of the city of Murcia, predominantly semi-arid environment with a combination of industrial and urban infrastructures and some areas of irrigated vegetation. CIAPA-IRIAF’s hives were located in a rural area characterized by extensive agricultural areas and small areas of riverside forest. Finally, the FAUNIA hives are located in an urban area, with landscape grounds. Drones were collected from controlled hives at an age of approximately 14–20 days, in sexually mature drones. It was unnecessary to submit this study to an ethics committee on animal experimentation since there is no regulation for experimentation or animal welfare on insects.

This study consisted of three experiments: Experiment 1 involved the study of optimal conditions of drone’s semen short-term conservation; Experiment 2 involved the effect of extender supplementation with catalase on seminal characteristics; and Experiment 3 required the gametic characterization by the sperm head morphometry assay.

### 2.2. Semen Collection and Sperm Variables Evaluation

In Experiment 1, seminal samples were collected from 30 drones from UMU and 50 from CIAPA-IRIAF. Semen collection was performed by everting the drone endophallus, using the technique described by [28]. One µL of semen per drone was collected using a Pasteur pipette with an elongated tip. From UMU drones, 3 pools were obtained, and from CIAPA-IRIAF drones, 5 pools were used (10 drones/pool), which were then grouped into 2 pools as homogeneous as possible according to sperm motility and viability. Sperm samples were collected during May and June. All pools were immediately diluted 1:1 (*v*/*v*) at room temperature with Kiev solution (sodium citrate 2.43 g, sodium bicarbonate 0.21 g, potassium chloride 0.04 g, glucose 0, 30 g, and 100 mL of H_2_O, pH 8.43, final osmolality 297 mOsm/kg) [16] in a 1.5 mL Eppendorf tube (Eppendorf Tubes^®^ 3810X, Hamburgo, Germany). The chemical reagents used were purchased from Panreac Química S.A. (Barcelona, Spain) and Sigma Chemical Co. Diluted samples were refrigerated (5–10 °C) and transported to the INIA-CSIC laboratory for analysis (24 h and 2 h, respectively, according to hive origin).

Sperm motility, acrosome status, sperm viability, and sperm abnormalities were initially evaluated for each group just after collection. The percentage of individual motile spermatozoa (motility) was noted, and the quality of motility (score) was evaluated using a scale of 0, the lowest, to 5, the highest. Morphological abnormalities and the percentage of spermatozoa with intact acrosomes were evaluated in samples stained with Hemacolor^®^ (Ref.: 1.11661, Merck KGaA, Darmstadt, Germany) by optical field microscopy at 40×. Propidium iodide and SYBR-14 (PI, Re.:L7011, Invitrogen. Thermo Fisher Scientific, Waltham, MA, USA) were used for sperm viability examination [29]. For this, 2 μL of SYBR-14 and 5 μL of semen diluted sample were placed in an Eppendorf tube containing 100 μL of HEPES medium (20 mM Hepes, 197 mM NaCl, 2.5 mM KOH and 10 mM glucose; pH 7.0, osmolality 400 mOsm/kg), and incubated at 5 °C for 10 min in the dark. One microliter of PI was then added, and the solution was incubated for 2 min at 5 °C. Samples were examined by epifluorescence microscopy (40×; wavelength 450–490 nm. Nikon Eclipse E200, Nikon Instruments Inc., Melville, NY, USA). Sperm showing a green color were considered alive, while red-colored sperm and cells with red and green color were considered dead (red coloration means that the membrane has been damaged and has lost its function). All analyses required observation of 200 cells.

The sperm variables were assessed under different incubation conditions: 5 °C, 15 °C, and 37 °C with 5% CO_2_) during 24 h, 48 h, and 72 h. We have compared the optimum temperature most commonly used in drone semen cooling (15 °C) with two extreme conditions: 37 °C (5% CO_2_) simulating body temperature at spermatheca, and liquid storage at 5 °C to reduce cellular metabolism. For incubation at 37 °C with 5% CO_2_ atmosphere, the incubator Labotect Inkubator C16 (Labor-Technik-Göttingen, Rosdorf, Germany) was used.

### 2.3. Supplementation with Catalase of the Drone Extender

In Experiment 2, 80 drones were captured from the FAUNIA hive and transported alive to our laboratory at room temperature. Once there, semen samples were collected and immediately diluted. Eight pools were obtained, of which 4 were diluted in Kiev solution (1/10, *v*/*v*) [16] and 4 in Kiev solution supplemented with 200 UI of catalase (1/10, *v*/*v*) (sodium citrate 2.43 g, sodium bicarbonate 0.21 g, potassium chloride 0.04 g, glucose 0.30 g, catalase 0.0059 g and 100 mL of H_2_O, pH 8.5, final osmolality 305 mOsm/kg). On the other hand, semen samples from 80 drones were collected at the CIAPA-IRIAF facilities, and the 8 pools were diluted in both solutions (4 pools/solution). All sperm samples were transported to our laboratory at 15 °C. In both cases, sperm motility and viability were evaluated initially after dilution with the extenders and after 3 h and 15 h at 15 °C.

### 2.4. Morphometric Analysis of the Sperm Head

In Experiment 3, 51 individual sperm samples were collected, corresponding to 17 drones from each origin (UMU, CIAPA-IRIAF, and FAUNIA hives). The samples belonged to the same colony for each origin. The individual samples were diluted 1:1 (*v*/*v*) at room temperature in Kiev solution. Smears were prepared by spreading 5 μL drops of diluted sperm samples onto glass slides and allowing them to air-dry. The samples were fixed and stained with Hemacolor^®^ (Merck KGaA, Darmstadt, Germany). Once stained and dried, all slides were sealed with Eukitt mounting medium (Panreac Química S.L.U., Barcelona, Spain) and coverslip. Smears were subjected to computerized morphometric analysis using Motic Image Advanced V.3.0 software (Motic Spain, S.L.U., Barcelona, Spain) and Motic BA 210 optical microscope (Motic Spain, S.L.U.) with a 100× oil immersion objective (bright field). Video signals were acquired using a 1SP 1.3 MP Moticam camera (Motic Spain S.L.) attached to the microscope and connected to a computer. Twenty-five spermatozoa per sample were randomly captured under the software’s manual acquisition mode [30]. Each sperm head was measured for length (L), width (W), area (A), perimeter (P), and length of acrosome (LAcros).

### 2.5. Statistical Analysis

Statistical analyses were performed using STATISTICA software for Windows v.12.0 (StatSoft, Inc., Tulsa, OK, USA). Data that did not follow a normal distribution were arcsine transformed. In Experiment 1, ANOVA-one way tested the effect of incubation temperatures on sperm viability. In Experiment 2, the effect of incubation time on seminal quality when using extender with and without catalase was also assessed by ANOVA. In Experiment 3, for each sperm head morphometric parameter, the mean and standard error (SEM) were calculated. Moreover, coefficients of variation (CVs) were calculated as the ratio between the standard deviation divided by the mean of each of the five morphometric parameters and expressed as a percentage. CVs within-drone were calculated as each drone’s individual variation in spermatic morphometry. This calculation was made as the ratio between the standard deviation divided by the mean for each of the five morphometric parameters of the drone and was expressed as a percentage. For the CVs between-drones, the sperm morphometric average of each drone was considered.

## 3. Results

### 3.1. Experiment 1

Drone seminal samples from UMU transported during 24 h after collection in refrigeration conditions (5–10 °C) showed, initially, low sperm motility (mean ± SEM) (11.7 ± 1.7%) and many abnormal shapes (60.7 ± 5.5%), whereas sperm viability (50.3 ± 8.0%) and acrosome integrity (72.0 ± 6.5%) were maintained. After 24 h of the initial evaluation, 25.3% and 54% of sperm viability was observed on samples stored at 5 °C and 15 °C, respectively. In addition, after 48 h and 72 h of the initial evaluation, the sperm viability values were reduced to 13–11% at 5 °C and 15.5–16% at 15 °C. The samples stored at 37 °C evaporated quickly, and it was not possible to perform this test.

Drone seminal samples from CIAPA-IRIAF transported during 2 h after collection in refrigeration conditions (5–10 °C) also showed low sperm motility and viability (mean ± SEM) (8.5 ± 6.5% and 30.5 ± 7.5%, respectively), whereas high values of acrosome integrity were also maintained (83.0 ± 8.0%). Incubations were conducted for 3 h and 15 h after initial evaluation at 5 °C, 15 °C, and 37 °C (5% CO_2_). After incubation for 3 h and 15 h, the greatest values of sperm motility and viability were found at 15 °C (Table 1).

### 3.2. Experiment 2

In seminal samples from FAUNIA, the incubation time negatively affected the sperm quality with both extenders (with or without catalase). This effect was observed in samples diluted in Kiev solution after 3 h of storage. However, when catalase was used, sperm motility and viability remained stable during the first hours, decreasing after 15 h (Figure 1 and Figure 2). In samples from CIAPA-IRIAF, there were no significant differences in sperm survival rate over time in any of the two diluents tested; the extender supplemented with catalase maintained stable sperm motility over time, but without significant differences (Figure 3).

### 3.3. Experiment 3

Hemacolor^®^ proved effective for staining drone sperm samples since it stained 100% of spermatozoa. A total of 25 sperm cells from each slide were captured and subsequently analyzed (n = 1275 over the entire set of semen samples) (Figure 4). Sperm head and acrosome morphometric parameters of drones obtained in this study were the following (mean ± SEM): L = 5.13 ± 0.00 µm; W = 0.85 ± 0.00 µm; A = 3.78 ± 0.02 µm^2^; *p* = 15.01 ± 0.01 µm; LAcros = 3.50 ± 0.02 µm. The between-drones CVs values were higher than the within-drone CVs, in all morphometric parameters and each of the origin groups (Table 2).

## 4. Discussion

The natural process of sperm storage in queen bees can be observed from two points of view. One corresponds to the initial sperm motility once inside the spermatheca, and the other is related to sperm longevity in this organ [31]. Based on these premises, we aimed to evaluate drone semen extenders capable of preserving sperm motility and viability over time. Previous studies have been conducted on semen composition in drones, in which ejaculate volume, mean concentration, motility, or sperm viability were evaluated [10,28,32]. Since the development of honey bee queens’ instrumental insemination, there has been interest in in vitro semen storage using different diluents and conditions [12,33]. Some studies reported that drone spermatozoa could be reversibly inactivated with an artificial buffer (pH: 7.19), which contained concentrations of Na^+^ and K^+^ similar to those found in the queen bees’ spermathecae. The high concentration of these cations may be responsible for the longevity of spermatozoa in an immobile state within the spermatheca [34].

Our first experiment used Kiev solution (pH = 8.43) as the drone semen diluent of choice [14]. The evaluation was carried out on diluted seminal samples transported at 5–10 °C and analyzed 24 h and 2 h post-collection, depending on the origin (UMU and CIAPA-IRIAF). Poor motility was observed, probably due to refrigeration conditions during transport and the time elapsed since collection. It has been shown that during prolonged semen storage in refrigeration, there is an increase in ROS production, a decrease in the amount of antioxidant substances, and in the sperm quality parameters [35,36]. A large number of abnormal shapes in spermatozoa were found in samples from UMU. These hives had been located in an area heavily treated with pesticides, indicating possible spermatogenic alterations [37]. Viability and acrosome status were preserved, in both cases. Subsequently, we tried to find out the optimal extender and the incubation conditions to preserve the functionality of the sperm over time. According to our results, the optimal temperature would be 15 °C, coinciding with those of Harbo and Williams [38], who showed that it is possible to store drone semen between 13 °C and 25 °C; however, sperm viability appears to be higher when storing between 12 °C and 16 °C, and the optimal conservation time would be 24 h. Curiously, sperm viability was slightly higher in UMU than in CIAPA-IRIAF, even though the UMU samples were analyzed 22 h later. Possibly, the genetic differences in each population and the influence of environmental conditions have led to this difference in sperm viability.

The addition of antioxidants to the extenders improves semen quality in drones [31,38,39], as they can prevent cell damage caused by oxidative stress [40]. Catalase is naturally present in the spermathecae of both mated and unmated queens, with higher levels in mated queens [41], as well as in the semen and spermatozoa of drones [42]. As we have indicated above, this enzyme requires iron as a cofactor to become active. The localization of catalase and its cofactor in reproductive tissues implies that catalase reduces oxidative damage to gametes and thus prolongs sperm survival [31,43]. Therefore, in the second experiment, we decided to add 200 IU of catalase to the initial Kiev solution. We were able to observe an improvement in seminal quality over the time of sperm exposure to this antioxidant, coinciding with what was reported by other authors [44,45,46].

There are numerous studies on seminal evaluation, extenders composition, and conditions storage in drone semen; however, few have also evaluated sperm head morphometry. To this end, we performed a third experiment to characterize these variables. The combination of sperm morphometry and sperm viability studies has been used to assess male fertility for a long time in various species [47,48,49,50]. Several systems have been developed to analyze the shape and size of spermatozoa automatically, achieving high precision, reliability, and objectivity. The Computer-Aided Semen-Morphometry Analysis Systems (CASA-Morph systems) have been standardized to analyze the semen of different species, mainly mammals, also establishing the most appropriate sampling and staining techniques [51]. CASA-Morph Systems have provided reliable results in morphometric studies and have been subsequently used to predict fertilization rates and sperm freezability [52]. However, the CASA-Morph systems are not yet developed enough to allow their use in drone sperm analysis, not even with other insects. The particular elongated shape of the drone spermatozoa, like a needle [28], makes these methods of sperm morphometry evaluation invalid for this species. Due to the morphological similarity of drone and bird spermatozoa, we have used Hemacolor^®^ staining and morphometric evaluation through computer-aided light microscopy, which had been successfully used in a previous study with birds (roosters and partridges) which also have phylliform spermatozoa [30]. Hemacolor^®^ was used as a valid staining technique, staining 100% of the drone sperm analyzed and revealing more details in the morphological structure of the spermatozoon. Gontarz et al. [11] analyzed the morphometric differences regarding the time of semen collection and the type of staining used. Their results do not agree with those obtained in our study, probably due to the use of different staining techniques and because the sperm head length includes the nucleus and acrosome together, and therefore, the parameters are higher. Our gametic characterization work turns out to be more extensive, defining essential measurements such as sperm head length (5.13 µm), width (0.85 µm), area (3.78 µm^2^), perimeter (15.01 µm), and acrosome length (3.50 µm) in drones, using Motic Image Advanced V3.0 software. We also analyzed the variability in the sperm head morphometry. Our results indicate that sperm differences within-drones (CV within-drones) are more significant than sperm differences between-drones (CV between-drones) for all morphometric parameters, as described in alpaca [53], dog [54], and ibex [55]. This finding suggests a high degree of sperm pleomorphism in drones, which could lead to an in-depth study of sperm subpopulations based on their morphometry in the future.

The variability in sperm head morphometry between areas may also be due to genetic variations in the populations studied. Thirty subspecies of honeybees (*Apis mellifera*) have been described worldwide [56]. Within Europe, 10 of these subspecies are found to be naturally distributed [57]. These subspecies are distinguished by a range of characteristics, including morphological, phenotypic, and molecular traits [58], which enable their classification into distinct evolutionary lineages. Within Europe, honeybee populations belonging to the African evolutionary lineage (A) are present in the southern region of the Iberian Peninsula, where *Apis mellifera iberiensis* is distributed, and to the Western European (M) lineage, which is found in Central Europe, where *A. m. mellifera* is distributed. Furthermore, the populations of these subspecies come into contact naturally [59], thus allowing the observation of a gradient of evolutionary lineage distribution. Specifically, two lineages are observed in the Iberian Peninsula: the aforementioned A in the southwest and M in the northeast. The samples included in this study belonged to the subspecies *Apis mellifera iberiensis*, but to different evolutionary lineages. The samples from CIAPA-IRIAF belonged to evolutionary lineage M, and those from the apiary of the UMU and FAUNIA to evolutionary lineage A.

In addition, we have to take into account that environmental factors may modify sperm head dimensions through disturbances in the chromatin package or by affecting acrosomal integrity. In CIAPA-IRIAF, bee populations may be more exposed to the effects of agricultural pesticides and fertilizers. On the other hand, populations in FAUNIA suffer from high pollution from the city, while those in UMU, located in the south of the Iberian Peninsula, may suffer greater thermal stress.

## 5. Conclusions

In conclusion, we can state that catalase improves drone sperm viability under storage conditions at 15 °C over time and that morphometry is a very useful tool to characterize drone sperm. Our results allow further studies of the honey bee colonies’ reproductive behavior, which can be affected by the use of toxic agents involved in the decline of pollinators.

## Figures and Tables

**Figure 1 animals-15-00672-f001:**
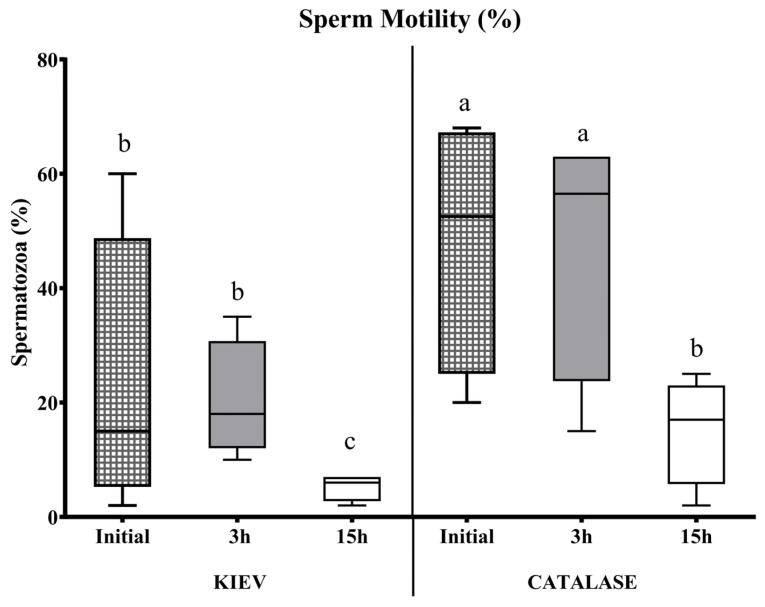
Sperm motility (%) evaluated at different incubation times in Kiev solution (KIEV) and Kiev solution supplemented with catalase (CATALASE), in drones from FAUNIA. The boxes spread from the 1st to the 3rd quartiles. Box plots show the median (horizontal line), and whiskers extend from the smallest up to the largest value. Different letters (a, b, c) indicate significant differences (*p* < 0.05).

**Figure 2 animals-15-00672-f002:**
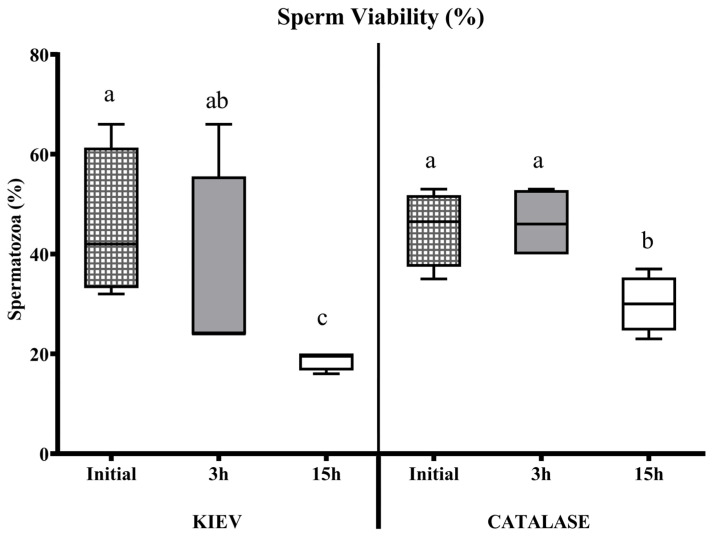
Sperm viability (%) evaluated at different incubation times in Kiev solution (KIEV) and Kiev solution supplemented with catalase (CATALASE), in drones from FAUNIA. The boxes spread from the 1st to the 3rd quartiles. Box plots show the median (horizontal line), and whiskers extend from the smallest up to the largest value. Different letters (a, b, c) indicate significant differences (*p* < 0.05).

**Figure 3 animals-15-00672-f003:**
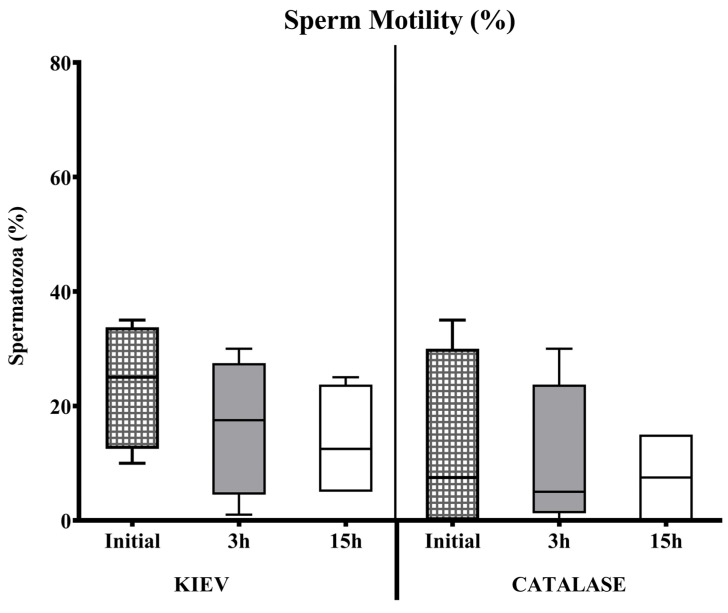
Sperm motility (%) evaluated at different incubation times in Kiev solution (KIEV) and Kiev solution supplemented with catalase (CATALASE), in drones from CIAPA-IRIAF. The boxes spread from the 1st to the 3rd quartiles. Box plots show the median (horizontal line), and whiskers extend from the smallest up to the largest value.

**Figure 4 animals-15-00672-f004:**
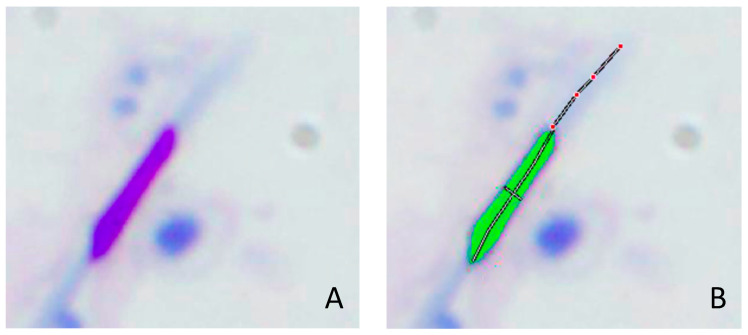
Fixed and stained sperm head with Hemacolor^®^ for morphometric analysis. (**A**): Stained sperm head. (**B**): Measurements: length, width, area, perimeter, and length of acrosome.

**Table 1 animals-15-00672-t001:** Sperm functional test under different incubation conditions in samples from CIAPA-IRIAF.

Sperm parameters after 3 h incubation	5 °C	15 °C	37 °C + 5% CO_2_
Motility (%)	1.5 ± 0.5 ^ab^	10.0 ± 5.0 ^b^	0.0 ± 0.0 ^a^
Score (0–5)	0.5 ± 0.0	0.8 ± 0.3	0.0 ± 0.0
Viability (%)	10.0 ± 8.0	12.5 ± 2.5	0.5 ± 0.5
Sperm parameters after 15 h incubation	5 °C	15 °C	37 °C + 5% CO_2_
Motility (%)	2.5 ± 2.5	6.5 ± 3.5	0.0 ± 0.0
Score (0–5)	0.25 ± 0.25	0.5 ± 0.0	0.0 ± 0.0
Viability (%)	5.0 ± 3.0	10.5 ± 4.5	4.5 ± 3.5

Mean ± SEM. Different superscripts (^a, b^) differ significantly (*p* < 0.05).

**Table 2 animals-15-00672-t002:** Within-drone and between-drones CVs of sperm cells stained with Hemacolor^®^ for 17 drones in each group according to their origin.

	CV (%)					
Origin		L	W	A	P	LAcros
UMU	Within-drone	4.18	13.03	11.03	3.78	9.42
	Between-drones	3.29	7.90	9.49	3.04	4.80
CIAPA-IRIAF	Within-drone	4.98	12.88	9.27	4.14	10.84
	Between-drones	2.04	5.82	6.79	1.96	4.43
FAUNIA	Within-drone	4.96	16.71	18.35	5.21	12.04
	Between-drones	3.79	13.95	11.72	4.52	5.39

A: area; CV: coefficients of variation; L: length; LAcros: length of the acrosome; P: perimeter; W: width.

## Data Availability

The datasets generated or analyzed during the current study are available from the corresponding author upon reasonable request.

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
