# Peer review of "Storage Conditions of Sperm Samples and Gametic Characterization by Sperm Head Morphometry in Drones (Apis mellifera)"

_animals, 2025, doi:10.3390/ani15050672_

Round 1
Reviewer 1 Report
Comments and Suggestions for Authors
Comments:
The authors evaluate the sperm function under 5 ºC, 15 ºC, and 37 ºC. Results showed that sperm viability was optimally maintained at 15 ºC (P˂0.05).Subsequently, catalase (200 UI) added is benefit to sperm viability and motility over time (P˂0.05). And then, sperm morphology was measured by staining technique.
Unfortunately, the authors should do more work to meet the title: Storage Conditions of Sperm Samples and Gametic Characteri- zation by Sperm Head Morphometry in Drones (Apis mellif- era).
Major comments:
As is mentioned in the discussion, the optimal temperature would be 15 ºC, coinciding with those of Harbo & Williams [38],that it is possible to store drone semen between 13 °C and 25 °C; however, sperm viability appears to be higher when storing between 12 °C and 16 °C. So, the authors should refine the experimental temperature instead of 5, 15, and 37 to obtain the optimal storage temperature.
Moreover, the authors also did not give an reference of the optimal storage time and the optimal concentration of catalase.
In the introduction, the authors describes many sperm-damaging factors, including parasites, pestcides, toxics and even climate change using a great deal of space. However, these factors are not the subject of the study.
Only one sentence mentioned that large number of abnormal shapes in spermatozoa was found in samples from UMU. These hives had been located in an area heavily treated with pesticides, indicating possible sper- matogenic alterations. Line 262
But, the article doesn't describe the environments of different sampling locations.
If authors want to explain these harmful factors, they should describe the sampling locations and explain the reasons for choosing Faculty of Veterinary Medicine of University from Murcia (UMU), Spain; Apicul- ture and Agro-environmental Research Center (CIAPA-IRIAF) from Marchamalo, Guada- lajara, Spain, and Faunistic Theme Park-FAUNIA from Madrid, Spain;
Minor comments:
The figures in the article need to be adjusted in terms of aesthetics.
Reference 10, 56, 57 58, 59 is unnecessary.
Simple summary should be improved.
Author Response
Comments:
The authors evaluate the sperm function under 5 ºC, 15 ºC, and 37 ºC. Results showed that sperm viability was optimally maintained at 15 ºC (P˂0.05).Subsequently, catalase (200 UI) added is benefit to sperm viability and motility over time (P˂0.05). And then, sperm morphology was measured by staining technique.
Unfortunately, the authors should do more work to meet the title: Storage Conditions of Sperm Samples and Gametic Characteri- zation by Sperm Head Morphometry in Drones (Apis mellif- era)
Major comments:
Comments 1: As is mentioned in the discussion, the optimal temperature would be 15 ºC, coinciding with those of Harbo & Williams [38],that it is possible to store drone semen between 13 °C and 25 °C; however, sperm viability appears to be higher when storing between 12 °C and 16 °C. So, the authors should refine the experimental temperature instead of 5, 15, and 37 to obtain the optimal storage temperature.
Response 1: First of all, we would like to thank you for reviewing this manuscript and for your time. All comments have been very appropriate and appreciated.
Short storage temperatures were chosen according to the following criteria: First, an optimal temperature was selected in the range recommended by the bibliography, between 12-16ºC. This optimum temperature for short-term storage was set at 15 degrees. After, two extreme temperatures were subsequently selected from this possible optimum: a temperature close to body temperature at spermatheca, 37ºC, and a temperature that reduces cellular metabolism to a minimum without freezing the medium (liquid storage at 5ºC). Furthermore, the store temperatures used, 5°C and 15°C, are the most commonly used in short-term storage in most species. In rams and goats, 5°C is used, in board it is better to store at 15°-17°C. In birds it is usually 5°C. We think that it is interesting to evaluate these extreme temperatures (chilled and putative temperature al spermatheca) including viability but also for motility variables. A brief explanation of the selected temperatures has been included in M&M (L130-135)
Comments 2: Moreover, the authors also did not give an reference of the optimal storage time and the optimal concentration of catalase.
Response 2: The aim of the project was not to indicate a specific storage time until which optimal semen quality conditions are maintained, but rather to show, using the three temperatures chosen, how semen quality decreases over the sampling time points. In this way, the reader will be able to assess the optimal storage time according to his needs. Likewise, the discussion section will indicate, based on the results obtained, what is the maximum storage time with good semen quality was 24h. (L260-265).
Regarding catalase, only a concentration of 200 IU was used. According to the results of experiment 2, a positive effect was produced on sperm motility and viability. In line 267-276 of the discussion section it is already reflected. After a search, we have only found one manuscript where catalase is added as an additive to a medium used for washing semen in drones. With concentrations different from those used in this work. (J Econ Entomol. 2014 Feb;107(1):47-53. doi: 10.1603/ec13159).
Comments 3: In the introduction, the authors describes many sperm-damaging factors, including parasites, pestcides, toxics and even climate change using a great deal of space. However, these factors are not the subject of the study.
Only one sentence mentioned that large number of abnormal shapes in spermatozoa was found in samples from UMU. These hives had been located in an area heavily treated with pesticideslake , indicating possible sper- matogenic alterations. Line 262
But, the article doesn't describe the environments of different sampling locations.
If authors want to explain these harmful factors, they should describe the sampling locations and explain the reasons for choosing Faculty of Veterinary Medicine of University from Murcia (UMU), Spain; Apicul- ture and Agro-environmental Research Center (CIAPA-IRIAF) from Marchamalo, Guada- lajara, Spain, and Faunistic Theme Park-FAUNIA from Madrid, Spain;
Response 3: Environmental pollutants and pesticides may modify sperm head dimensions through disturbances in the chromatin package or by affecting acrosomal integrity, and sperm morphometry was an objective of the current study.. In this manuscript, the effect of these factors on sperm quality and morphometry is not intended to be analyzed, but rather to establish the first knowledge of the morphometry of different drone populations for later studies. On the other hand, the short-term storage protocol is intended to be optimized for future studies of semen quality in different populations with different environmental conditions. Indications have been made (L309-311)
The three selected centers have been chosen because they are healthy, controlled hives, no pest control treatments are used, they have not maintained any exchange of honeycombs or specimens, and they are located in areas far from each other. The three places are not described in more detail as it is not considered necessary for the objectives of the manuscript. Explaining the effects of environmental factors on semen quality would require a more complete and complex experiment.
Minor comments:
Comments 4: The figures in the article need to be adjusted in terms of aesthetics.
Response 4: Modifications to the figures have been made
Comments 5: Reference 10, 56, 57 58, 59 is unnecessary.
Response 5: References 10, 56 and 59 have been removed. Reference 57 (now XX) has been retained as an example reference for CASA analysis and 58 (now XX) as an example for predict fertilization rates and sperm freezability. Following the recommendations of the editor and other reviewers, other self-references throughout the manuscript have been removed. Old reference numbers removed: 5, 10, 23, 26, 30, 56, 59 and 62
Comments 6: Simple summary should be improved.
Response 6: Simple summary has been improved.
Reviewer 2 Report
Comments and Suggestions for Authors
Honey bees are important pollinators, but their relevance depends on the landscape. In areas where they are not native or with diverse plant species, they are not always the primary pollinators. Replace 'main pollinators' with 'important pollinators' or similar, as it is not accurate.
Line104- 106. it is relevant to consider whether the "pools" come from the same colony or different colonies, as this can affect the results and interpretation of the experiments for several reasons: Genetic diversity, Environmental and nutritional conditions and Experimental control. Due those points, if the "pools" include drones from different colonies, it is important to mention this explicitly. This could lead to misinterpretation of the differences as variations within (within drones) and between (between drones). If the pools come from different colonies, what we might actually be observing are differences between colonies within a pool, while the between differences could be more related to variations in sites or management conditions.
Line 182-Justify why those temperatures were used. Particularly the 37°C
Figures 1, 2,3- include the statistical significances and improve the plots. For me, seems to basic use excel, can be more attractive.
Line 294- This needs a reference (A reference that states that falcon and drone sperm are similar). If it doesn't exist, avoid referencing falcon sperm; instead, simply mention that you are testing the method or something similar. Drawing comparisons between birds and insects seems out of context.
Author Response
REVIEWER 2
Comments 1: Honey bees are important pollinators, but their relevance depends on the landscape. In areas where they are not native or with diverse plant species, they are not always the primary pollinators. Replace 'main pollinators' with 'important pollinators' or similar, as it is not accurate.
Response 1: Thank you very much for taking the time to review this manuscript. Please find the detailed responses below and the corresponding corrections in the re-submitted files.
We replace 'main pollinators' with 'important pollinators'. (L42)
Comments 2: Line104- 106. it is relevant to consider whether the "pools" come from the same colony or different colonies, as this can affect the results and interpretation of the experiments for several reasons: Genetic diversity, Environmental and nutritional conditions and Experimental control. Due those points, if the "pools" include drones from different colonies, it is important to mention this explicitly. This could lead to misinterpretation of the differences as variations within (within drones) and between (between drones). If the pools come from different colonies, what we might actually be observing are differences between colonies within a pool, while the between differences could be more related to variations in sites or management conditions.
Response 2: Thanks for this comment, it's very interesting. The drones were caught at the exits of the hives. Each pool was made with drones from the same hive. We therefore consider that the "pools" include drones from the same colony. In any case, of experiment 3, the morphometric analyses were done by individual, not by "pool". Therefore, a comparison can be made between and within drones. Modifications have been made L149-150
Comments 3: Line 182-Justify why those temperatures were used. Particularly the 37°C.
Short storage temperatures were chosen according to the following criteria: First, an optimal temperature was selected in the range recommended by the bibliography, between 12-16ºC. This optimum temperature for short-term storage was set at 15 degrees. After, two extreme temperatures were subsequently selected from this possible optimum: a temperature close to body temperature at spermatheca, 37ºC, and a temperature that reduces cellular metabolism to a minimum without freezing the medium (liquid storage at 5ºC). Furthermore, the store temperatures used, 5°C and 15°C, are the most commonly used in short-term storage in most species. In most mammal species the optimum short-term storage is at 5°C. In birds where sperm are also fusiform, like drones, the short-term conservation is too at 5°C. Meanwhile, the temperature of 15°C was selected in board because it is a species with sperm sensitive to low temperatures.L130-135
Comments 4: Figures 1, 2,3- include the statistical significances and improve the plots. For me, seems to basic use excel, can be more attractive.
Response 4: Modifications to the figures have been made. The statistical significances has been include
Comments 5: Line 294- This needs a reference (A reference that states that falcon and drone sperm are similar). If it doesn't exist, avoid referencing falcon sperm; instead, simply mention that you are testing the method or something similar. Drawing comparisons between birds and insects seems out of context.
Response 5: First of all, I would like to apologize for a mistake in this sentence. The reference mentioned (previously numbered 35, now 30) refers to roosters and partridges, not falcons. The error has been corrected (293-294). Even so, in this reasoning what is being compared is the morphology of the sperm, the external physical appearance, which can be observed simply under a microscope. These groups: drones, rooster and partridges (and other birds) present a fusiform sperm, like a needle, very different from many mammals, with a more globose shape. Considering this perspective, simply from the physical aspect, we believe that the comparison between birds and insects is appropriate for the study of the Hemacolor technique using Motic Image Advanced V3.0 software. Furthermore, the results of this study support that the stain used in birds is valid for the drone. This paragraph does not assess the functional implications of the spermatozoon related to its appearance or the physical-chemical relationships between the spermatozoon and the external environment, which may also vary depending on its appearance.
Reviewer 3 Report
Comments and Suggestions for Authors
This paper describes the effect of different storage durations of honeybee sperm in vitro and the influence of catalase addition on sperm viability and motility, while also observing the sperm head morphology. Overall, the paper conducts some experiments and obtains some results that are somewhat meaningful for reference, but due to experimental design flaws and limited data, the conclusions are not solid. Additionally, the data presentation in the key figures is unsatisfactory, with poor clarity and a lack of important statistical results. Therefore, I do not recommend publishing this paper. The specific issues are as follows:
1. The age of drones is indeed a very critical factor influencing semen quality and sperm development. Unfortunately, this factor has not been explicitly considered or addressed in this study and experimental design. This omission could impact the interpretation of experimental results and the reliability of the conclusions. Drones typically require approximately 10–14 days to reach sexual maturity. Before this period, their sperm may not be fully developed, and semen quality (e.g., motility, acrosome integrity, and morphology) may be below optimal levels. Drone age also affects their endocrine status, which in turn influences sperm production, storage, and the levels of cofactors (e.g., antioxidants) in the semen.
2. The experiment testing sperm viability under different storage times and the effects of catalase addition clearly lacks sufficient time points. Only two time points were set, which is obviously inadequate. Multiple time points should be included within the 0-48 hour range, and continuous fluctuation curves should be plotted. This would provide more confidence in the experimental results.
3. Figures 1-3 are drawn very roughly and lack significant markers in all the graphs, making it impossible to determine whether the conclusions are correct.
4. Sperm head and acrosome structure from bees of different origins were tested, but the developmental status of the sperm was not considered. This could lead to variability among experimental groups and undermines the reliability of the conclusions.
5. The paper contains excessive self-citation. It is acceptable to reference studies on sperm of other species by the authors, however, the references seem to be excessive in this paper.
Author Response
This paper describes the effect of different storage durations of honeybee sperm in vitro and the influence of catalase addition on sperm viability and motility, while also observing the sperm head morphology. Overall, the paper conducts some experiments and obtains some results that are somewhat meaningful for reference, but due to experimental design flaws and limited data, the conclusions are not solid. Additionally, the data presentation in the key figures is unsatisfactory, with poor clarity and a lack of important statistical results. Therefore, I do not recommend publishing this paper. The specific issues are as follows:
Comments 1. The age of drones is indeed a very critical factor influencing semen quality and sperm development. Unfortunately, this factor has not been explicitly considered or addressed in this study and experimental design. This omission could impact the interpretation of experimental results and the reliability of the conclusions. Drones typically require approximately 10–14 days to reach sexual maturity. Before this period, their sperm may not be fully developed, and semen quality (e.g., motility, acrosome integrity, and morphology) may be below optimal levels. Drone age also affects their endocrine status, which in turn influences sperm production, storage, and the levels of cofactors (e.g., antioxidants) in the semen.
Response 1: All authors would like to thank you for the time and dedication you have taken to review and prepare the comments. We have carefully read and analyzed all the suggestions and we hope that our responses are satisfactory to you.
Indeed, as you indicate, the age and maturity of the drone is very important in the quality of the semen, as occurs in all species. Taking into account the experience of several authors, the age and maturity of the drones were taken into account when designing the experiment. The hives were controlled and the drones were collected in each population at the appropriate time. The approximate age of the drones was 14-20 days, and it was ensured that they had made flights. In the M&M section the approximate age of the drones was included ( L90-92)
Comments 2. The experiment testing sperm viability under different storage times and the effects of catalase addition clearly lacks sufficient time points. Only two time points were set, which is obviously inadequate. Multiple time points should be included within the 0-48 hour range, and continuous fluctuation curves should be plotted. This would provide more confidence in the experimental results.
Response 2: Indeed, the more sampling points are carried out in the experiment, the more robust the results will be. However, in most short-term semen storage studies, semen quality control points are usually carried out every 24 hours, since what is sought is how many days the semen can be preserved under these conditions with the best possible quality. Therefore, the goal is not to store hours but rather days. Unfortunately, and as we believed when designing the experiment, the semen could not be preserved for many more days. For this reason, the experiment with catalase was adjusted to two control point sampling, at 3 h and at 15 h.
Comments 3. Figures 1-3 are drawn very roughly and lack significant markers in all the graphs, making it impossible to determine whether the conclusions are correct.
Response 3: Modifications to the figures have been made
Comments 4. Sperm head and acrosome structure from bees of different origins were tested, but the developmental status of the sperm was not considered. This could lead to variability among experimental groups and undermines the reliability of the conclusions.
Response 4: As we indicated in the response to comment 1, the age and maturity of the drones were taken into account, within the margins of variation that may exist in each hive. Therefore, the morphometric parameters can be compared considering the variable "maturity of the spermatozoa" as fixed.
Comments 5. The paper contains excessive self-citation. It is acceptable to reference studies on sperm of other species by the authors, however, the references seem to be excessive in this paper.
Response 5: We have reviewed the manuscript and removed many references by the authors. Old reference numbers removed: 5, 10, 23, 26, 30, 56, 59 and 62
Reviewer 4 Report
Comments and Suggestions for Authors
The work presented here addresses the problem of declining sperm quality in drones, and thus the difficulty in obtaining fertilization and offspring by the queen. The problem addressed is an important issue in the study of the declining bee population as a result of human activities. The importance of bees to ecosystems is enormous, so any attempt to allow the conservation of their population is important.
The layout of the paper is correct, but I have a few questions for the Authors.
1) Why the choice of 15 degrees Celsius for storage conditions? In the case of sperm drones, the more common storage temperature for sperm in the literature is 12 degrees.
2) Why was 15 hours chosen as the storage time? What is the rationale for analyzing sperm motility and viability after 15h? Usually in mammalian semen studies, short-term storage time is considered, in this case 1 or 3h, or long-term storage (daily cycles) of 24h, 48h, 72h.
3) What was the viability of semen from UMU stored at 37 degrees for 24 hours? I couldn't find any data in the description, unless this time has already caused the sample to evaporate?
4) What was the reason for the much lower "initial" viability of semen from CIAPA-IRIAF compared to UMU?
5) The authors found a positive effect of catalase on sperm viability and morphology, but have you considered or tested the effect of DMSO on drone sperm? Or other agents that have been successfully used to cryopreserve drones' sperm, such as the addition of chicken egg yolk, which could also have a positive effect on short-term storage.
Author Response
The work presented here addresses the problem of declining sperm quality in drones, and thus the difficulty in obtaining fertilization and offspring by the queen. The problem addressed is an important issue in the study of the declining bee population as a result of human activities. The importance of bees to ecosystems is enormous, so any attempt to allow the conservation of their population is important.
The layout of the paper is correct, but I have a few questions for the Authors.
Comments 1) Why the choice of 15 degrees Celsius for storage conditions? In the case of sperm drones, the more common storage temperature for sperm in the literature is 12 degrees.
Response 1: Estimated reviewer, I would like to thank you in advance for your dedication and effort in reviewing our manuscript. All your comments have been taken into account to improve this manuscript.
Twe storage temperatures were chosen according to the following criteria: First, an optimal temperature was selected in the range recommended by the bibliography, between 12-16ºC. This optimum temperature for short-term storage was set at 15 degrees. After, two extreme temperatures were subsequently selected from this possible optimum: a temperature close to body temperature at spermatheca, 37ºC, and a temperature that reduces cellular metabolism to a minimum without freezing the medium (liquid storage at 5ºC). Furthermore, the store temperatures used, 5°C and 15°C, are the most commonly used in short-term storage in most species. In most mammal species the optimum short-term storage is at 5°C. In birds where sperm are also fusiform, like drones, the short-term conservation is too at 5°C. Meanwhile, the temperature of 15°C was selected in board because it is a species with sperm sensitive to low temperatures.
A brief note of this criterion was included in M&M (L 130-135)
Comments 2) Why was 15 hours chosen as the storage time? What is the rationale for analyzing sperm motility and viability after 15h? Usually in mammalian semen studies, short-term storage time is considered, in this case 1 or 3h, or long-term storage (daily cycles) of 24h, 48h, 72h.
Response 2: Experiment 1 was carried out in two parts, a first sampling at the UMU, with incubation times of 24, 48 and 72 hours. Given the low semen quality obtained after 24 hours, the second sampling, at CIAPA-IRIAF, was designed at 3 and 15 hours to record the quality, at shorter times than 24 h.
Comments 3) What was the viability of semen from UMU stored at 37 degrees for 24 hours? I couldn't find any data in the description, unless this time has already caused the sample to evaporate?
Response 3: Effectively, the samples stored at 37 ºC evaporated quickly, and it was not possible to perform this test. (Line 186)
Comments 4) What was the reason for the much lower "initial" viability of semen from CIAPA-IRIAF compared to UMU?
Response 4: This issue also caught our attention when we analyzed the results. Firstly, the viability of both is low, something we did not expect (UMU 50.3±8.0%, CIAPA-IRIAF 30.5±7.5%) Sperm viability was slightly higher in UMU than in CIAPA-IRIAF, even though the UMU samples were analyzed 22 hours later. Possibly the genetic differences of each population, and the influence of environmental conditions have led to this difference in sperm viability. A more extensive study where a greater number of pools are collected per area and where the quality can be observed at the time (0-1h), could confirm this assumption. We have included this interesting observation in the Discussion section of the revised version (L261-264).
Comments 5) The authors found a positive effect of catalase on sperm viability and morphology, but have you considered or tested the effect of DMSO on drone sperm? Or other agents that have been successfully used to cryopreserve drones' sperm, such as the addition of chicken egg yolk, which could also have a positive effect on short-term storage.
Response 5: One of the aims of the manuscript is to reduce free radicals by using antioxidants. There are a large number of compounds with antioxidant effects that have been used in numerous studies, including catalase. It is proven that the supplementation of extenders with catalase improves sperm parameters in several species. On the other hand, DMSO is a compound with antioxidant action but is mainly used, due to its chemical properties, as a cryoprotectant in freezing extenders. Due to its toxic effect, it is not frequently used in refrigeration extenders.
Round 2
Reviewer 1 Report
Comments and Suggestions for Authors
The author has made improvements to the article, but there are still some questions remaining.
Comment 1
The results of Experiment 1 and Experiment 2 based on drones from different experimental apiaries are different. Additionally, the discussion also mentions the possibility of environmental influences. Therefore, the source of the sperm samples is crucial to this experiment. The details of the apiary and samples are essential for the manuscript.
Comment 2
The discussion section poorly explores the diversity in results deriving from samples of the three apiaries.
Other comments:
Climate change should not be a keyword.
The figures need to further refine.
Author Response
The author has made improvements to the article, but there are still some questions remaining.
Response: First of all, we would like to thank the reviewer for all the comments and contributions that are greatly improving this manuscript. We continue working on improving it in response to these new comments.
Comment 1
The results of Experiment 1 and Experiment 2 based on drones from different experimental apiaries are different. Additionally, the discussion also mentions the possibility of environmental influences. Therefore, the source of the sperm samples is crucial to this experiment. The details of the apiary and samples are essential for the manuscript.
Response 1: In the M&M section, specifications for the location of the hives and the biogeographic environment have been included. (L86-96).
Comment 2
The discussion section poorly explores the diversity in results deriving from samples of the three apiaries.
Response 2: To improve the discussion of the diversity of results from the three apiaries, we have delved deeper into the study of the bee populations used in this experiment.
Within Europe, honeybee populations belonging to the African evolutionary lineage (A) are present in the southern region of the Iberian Peninsula, where Apis mellifera iberiensis is distributed, and to the Western European (M) lineage, which is found in Central Europe, where A. m. mellifera. The distribution of the Western European subspecies (A. m. mellifera) is naturally dispersed, while the Eastern European subspecies (A. m. ligustica) is found in Italy and central-eastern European countries (A. m. carnica), and the Hellenic Peninsula (A. m. cecropia). Furthermore, the populations of these subspecies come into contact naturally (Muñoz and De la Rúa 2021), thus allowing the observation of a gradient of evolutionary lineage distribution. Specifically, two lineages are observed in the Iberian Peninsula: the aforementioned A in the southwest and M in the northeast.
The samples included in this study belonged to the subspecies Apis mellifera iberiensis, but to different evolutionary lineages (Sánchez-Aroca et al. unp.). The samples from CIAPA-IRIAF belonged to evolutionary lineage M, and those from the apiary of the University of Murcia and FAUNIA to evolutionary lineage A.
On the other hand, a reflection has been included on the most important environmental factors that differentiate the three zones: pollution in FAUNIA, agricultural management in CIAPA-IRIAF or heat stress in UMU.
Differences due to drone origin and environmental effects have been included in the discussion. (L 316-339) New references have been included in this paragraph
Other comments:
Climate change should not be a keyword.
Response 3: The word “climate change” has been removed
The figures need to further refine.
Response 4: The three figures have been completely modified. In the new version, the "boxplot" graphs have been used, where the results are better visualized. We can see the median, quartiles and extremes of the data.
